# *PRKAG2* Syndrome: Clinical Features, Imaging Findings and Cardiac Events

**DOI:** 10.3390/biomedicines13030751

**Published:** 2025-03-19

**Authors:** Maria Sudomir, Przemysław Chmielewski, Grażyna Truszkowska, Mariusz Kłopotowski, Mateusz Śpiewak, Marta Legatowicz-Koprowska, Monika Gawor-Prokopczyk, Justyna Szczygieł, Joanna Zakrzewska-Koperska, Mariusz Kruk, Jolanta Krzysztoń-Russjan, Jacek Grzybowski, Rafał Płoski, Zofia T. Bilińska

**Affiliations:** 1Unit for Screening Studies in Inherited Cardiovascular Diseases, Cardinal Stefan Wyszyński National Institute of Cardiology, 04-628 Warsaw, Poland; msudomir@ikard.pl (M.S.); pchmielewski@ikard.pl (P.C.); 2Molecular Biology Laboratory, Department of Medical Biology, Cardinal Stefan Wyszyński National Institute of Cardiology, 04-628 Warsaw, Poland; gtruszkowska@ikard.pl (G.T.); jkrzyszton@ikard.pl (J.K.-R.); rploski@ikard.pl (R.P.); 3Department of Cardiology and Interventional Angiology, National Institute of Cardiology, 04-628 Warsaw, Poland; mklopotowski@ikard.pl; 4Cardiomyopathy Outpatient Clinic, Cardiac Arrhythmia Center, Cardinal Stefan Wyszyński National Institute of Cardiology, 04-628 Warsaw, Poland; 5Magnetic Resonance Unit, Department of Radiology, National Institute of Cardiology, 04-628 Warsaw, Poland; mspiewak@ikard.pl; 6Department of Pathomorphology, National Institute of Geriatrics, Rheumatology and Rehabilitation, 02-637 Warsaw, Poland; 7Department of Cardiomyopathy, National Institute of Cardiology, 04-628 Warsaw, Poland; mgawor@ikard.pl (M.G.-P.); jszczygiel@ikard.pl (J.S.); jgrzybowski@ikard.pl (J.G.); 81st Department of Arrhythmia, National Institute of Cardiology, 04-628 Warsaw, Poland; jzakrzewska@ikard.pl; 9Coronary Artery and Structural Diseases Department, National Institute of Cardiology, 04-628 Warsaw, Poland; mkruk@ikard.pl; 10Department of Medical Genetics, Medical University of Warsaw, 02-106 Warsaw, Poland

**Keywords:** cardiomyopathy, hypertrophic, genetics, next generation sequencing, cardiac magnetic resonance, endomyocardial biopsy, *PRKAG2*

## Abstract

**Background/Objectives:** *PRKAG2* syndrome (PS) is a rare genocopy of hypertrophic cardiomyopathy (HCM). Our goal was to expand knowledge about PS by analyzing patient clinical, imaging, and follow-up data. **Methods**: The study included carriers of likely pathogenic or pathogenic *PRKAG2* variants identified in the years 2011–2022. Cardiac involvement was assessed by electrocardiography, echocardiography, cardiac magnetic resonance imaging, and endomyocardial biopsy (EMB). We recorded concomitant diseases and cardiac events, including the implantation of electronic cardiac devices, arrhythmia, heart failure (HF), and death. **Results**: Seven patients from four families (median age 43 years) with *PRKAG2* variants: Phe293Leu, Val336Leu, Arg302Gln, and His530Arg were included. At the first evaluation, 3 carriers were in New York Heart Association (NYHA) functional class II–III, while the remaining were in NYHA class I. Left ventricular hypertrophy (LVH) was present in 5 patients; 2 had ventricular pre-excitation, one was in atrial flutter and pacemaker-dependent; 2 had bradycardia. Two female carriers had concomitant chronic renal disease. In the EMB of one of the patients, staining for glycogen deposits was positive. Furthermore, we provide a link between the Val336Leu *PRKAG2* variant and autophagy identified on EMB. After a median follow-up of 13.1 years, 6 carriers had LVH, 3 required admission for HF, and 1 had sustained ventricular tachycardia with subsequent cardioverter defibrillator implantation, and despite this, died suddenly; there were two de novo pacemaker implantations due to symptomatic bradycardia. **Conclusions**: PR is a distinctive disorder with an early onset of arrhythmic events, often leading to HF.

## 1. Introduction

The *PRKAG2* syndrome (PS) is a rare, inherited disorder transmitted in an autosomal dominant manner, characterized by cardiac hypertrophy, pre-excitation features, and arrhythmias [1,2]. Diagnosing PS is important, as the syndrome is associated with a prevalence of early onset sino-atrial or conduction disease, advanced heart failure (HF), and potentially lethal arrhythmic events [2,3,4]. The most common genetic causes of HCM are pathogenic variants in eight genes encoding sarcomere proteins: *MYH7, MYBPC3, TPM1, TNNT2, MYL2, MYL3, TNNI3,* and *ACTC1*, of which variants in *MYBPC3* and *MYH7* account for ~75% of identified pathogenic variants [5,6]. Apart from the aforementioned eight sarcomeric genes, rare variants in more than a dozen additional genes were described; however, their evidence for HCM causality is not definitive. Another important aspect of genetic testing in HCM is HCM genocopies, such as metabolic storage diseases, Noonan syndrome, or mitochondrial cardiomyopathies, which mimic sarcomeric HCM but are caused by pathogenic variants in other genes, e.g., *GLA*, *TTR*, *FHL1*, *LAMP2*, *PTPN11*, and *PRKAG2* (Online Mendelian Inheritance in Man (OMIM) #602743) [6]. *PRKAG2* gene encodes a non-catalytic Gamma 2 subunit of adenosine-5′-monophosphate-activated protein kinase (AMPK). AMPK acts as a sensor of energy deficiency in the cell. In the case of oxidative stress in cardiomyocytes, it increases the production of ATP (stimulating the consumption of glucose and fatty acids) and limits its consumption (by limiting anabolic processes) [7]. AMPK expression is strongest in the heart muscle, and the myocardial histological hallmark of *PRKAG2* variants is widespread intracellular vacuolization, interstitial fibrosis, and deposition of amylopectin—a glycogen-like substance, which is a morphological indicator of impaired function of this enzyme [8,9]. Several transgenic murine models of *PRKAG2* variants were developed. They all uniformly reflected the disease observed in humans, with cardiac hypertrophy, dilatation, decreased contractility of the left ventricle found in echocardiography, and pre-excitation features in surface ECG [10]. Since the identification of the first pathogenic variant in 2001 [1], more than 30 *PRKAG2* pathogenic variants responsible for PS have so far been identified. Our goal was to report the clinical, imaging, and follow-up data of patients genetically diagnosed with PS at the National Institute of Cardiology to expand the knowledge of the clinical course of this syndrome and to provide a link between a *PRKAG2* variant and autophagy identified in myocardial biopsy [11].

## 2. Materials and Methods

### 2.1. Genetic Testing

The study cohort consisted of carriers of pathogenic or likely pathogenic *PRKAG2* variants, which were identified at the National Institute of Cardiology, Warsaw, using next-generation sequencing (NGS). Genetic testing using different gene panels was offered to index patients referred to the Institute between the years 2011 and 2022 with a diagnosis of hypertrophic cardiomyopathy (HCM) or restrictive cardiomyopathy (RCM). *PRKAG2* pathogenic variants were detected by whole exome sequencing (WES) in the first proband; a commercial panel consisting of 174 genes was used in the second and fourth probands (TruSight Cardio (TSC), Illumina, San Diego, CA, USA), and the TruSight One (TSO) sequencing panel (Illumina, San Diego, CA, USA) was used in the third proband. Sequencing was performed on HiSeq or MiSeq Dx (Illumina, San Diego, CA, USA). NGS results were inspected for rare (minor allele frequency < 0.001 for dominant or <0.05 for recessive disorders), protein-coding, or splicing variants in genes with definitive, strong, or moderate evidence for HCM or HCM genocopies/intrinsic cardiomyopathies according to ClinGen [6] and EMQN recommendations [12]. Identified variants were evaluated according to the American College of Medical Genetics and Genomics criteria [13] with the use of online tools: GeneBe.net https://genebe.net, VarSome https://varsome.com, Franklin by Genoox https://franklin.genoox.com (accessed on 28 November 2024), and literature search. In all probands, a three-to-four-generation pedigree was drawn, and the family history of cardiomyopathy and other diseases was obtained. Subsequent cascade screening was offered to all probands’ relatives, and identified variants were followed up in with Sanger sequencing using BigDye Terminator v3.1 or v1.1 Cycle Sequencing Kit (Life Technologies, Carlsbad, CA, USA) according to the manufacturer’s instructions and the 3500xL or 3130xl Genetic Analyzer (Life Technologies, Carlsbad, CA, USA). The results were analyzed with Variant Reporter 1.1 Software (Life Technologies, Carlsbad, CA, USA).

The study was conducted according to the guidelines of the Declaration of Helsinki and was approved by the Bioethics Committee of the National Institute of Cardiology, Warsaw, Poland. All participants gave written informed consent.

### 2.2. Clinical Evaluation and Follow-Up

Medical data of probands and relatives were retrospectively collected, including baseline clinical information from the first documented visit to the Institute, prior medical records, and follow-up data. We analyzed the baseline data, comprising medical history, clinical examination, standard 12-lead electrocardiography (ECG), two-dimensional Doppler echocardiography, cardiac magnetic resonance, and ambulatory ECG monitoring.

The patients were considered clinically affected by PS if they met one or more of the following criteria: otherwise, unexplained LVH (maximal LV thickness ≥ 13 mm), advanced conduction system disorders, sustained ventricular tachycardia, supraventricular tachycardia, and ECG abnormalities (pre-excitation, conduction system disease, repolarization abnormalities). Ventricular pre-excitation was diagnosed with the combined presence of a short PR interval ≤ 120 ms and a widened QRS complex ≥ 110 ms or with an abnormal delta wave [9]. The hypokinetic phase of PS was diagnosed according to the definition of hypokinetic HCM with a left ventricular ejection fraction (LVEF) < 50% [14,15]. Right ventricular endomyocardial bioptate sections, obtained to exclude amyloid in proband 3, were stained with hematoxylin-eosin and additionally with periodic acid Schiff (PAS) to reveal any glycogen deposits. Biopsy material was examined by light microscopy. Previously published data [11] on the results of endomyocardial biopsy in proband 1 were reviewed and linked to the *PRKAG2* variant for the first time.

Details of clinical events during follow-up were also collected. Events were considered as follows: de novo pacemaker implantation, sustained ventricular tachycardia, hospitalization for HF, myocardial infarction (MI), and sudden cardiac death. Death was considered sudden when it happened unexpectedly due to cardiac causes occurring within 1 h of the onset of symptoms. The end of observation was 30 September 2024.

The results for the categorical variables were presented as numbers and percentages, and for continuous variables, as median and range.

## 3. Results

### 3.1. Genetics

When examining patients with HCM/RCM (*n* = 279), pathogenic/likely pathogenic variants of the *PRKAG2* gene were found in 4 (1.4%) probands. Family testing revealed another three carriers of these variants. All variants were heterozygous. They were classified as pathogenic or likely pathogenic according to the ACMG criteria. Details concerning identified *PRKAG2* variants are summarized in Table 1, and the pedigrees of the families are shown in Figure 1. No other protein-coding or splice site variants with MAF < 0.001 were found in our probands in HCM-associated genes.

### 3.2. Clinical Picture

The disease was diagnosed in 2 men and 5 women at a median age of 43 years (range 29–66 years). The onset of the disease was most often characterized by heart palpitations (*n* = 4), dizziness (*n* = 2), and chest pain (*n* = 3). Dyspnea on exertion and limited exercise tolerance in daily activities were reported by 3 patients. One male patient with the Val336Leu variant suffered from a stroke, and his mother, a 66-year-old female carrier, had only mild ECG abnormalities, with no signs of cardiac hypertrophy. The Arg302Gln variant carrier experienced paroxysmal supraventricular tachycardia at the onset of symptoms that did not respond to adenosine and amiodarone and resolved after cardioversion. Atrioventricular parahisian accessory pathway was identified in this patient but was not treated due to the risk of His bundle damage. His mother and her sister had presented for initial evaluation with marked bradycardia.

None of the patients reported symptoms of skeletal muscle weakness, and serum creatine kinase activity values were normal. The clinical characteristics of all identified *PRKAG2* variant carriers are summarized in Table 2.

At first evaluation, one patient with the His530Arg variant was in atrial flutter (AFl), pacemaker-dependent, and the remaining 6 were in sinus rhythm; 2 of them had bradycardia (Figure 2). In 2 patients, pre-excitation features were observed, and the delta wave was found in one of them. In 4 patients, intraventricular conduction disorders were found in the form of isolated right bundle branch block (*n* = 1), right bundle branch block with left anterior fascicular block (*n* = 1), or non-specific intraventricular conduction disorders (*n* = 2). Furthermore, the patient with the His530Arg variant had a pacemaker implanted at the age of 21 years due to the disease of 2 nodes: sinus and atrioventricular, accompanied by left bundle branch block. In total, conduction disease was found in 5 patients.

In 4 carriers, ECG criteria for LVH were met; 5 patients had myocardial thickness ≥ 13 mm on echocardiography, 2 patients had LVEF ≤ 50%, and 2 had elevated right ventricular systolic pressure. Two carriers with His530Arg and Phe293Leu had concomitant chronic renal disease; long-lasting diabetes mellitus was also present in the latter.

### 3.3. Endomyocardial Biopsy

In two patients (proband 1 and proband 3), a myocardial biopsy was performed. In proband 1, long before the era of genetic testing, vacuolization in cardiomyocytes was observed by light microscopy, and an ultrastructural analysis revealed dramatic alterations of myocyte architecture in approximately 20% of cardiomyocytes, including the presence of autophagic vacuoles, glycogen granules, myelin-like structures, cytoplasmic remnants, and mitochondria [11]. In proband 3, on light microscopy, standard hematoxylin-eosin staining revealed hypertrophied myocytes with enlarged pleomorphic nuclei, and PAS staining was positive for glycogen accumulation in the cardiomyocytes (Figure 3).

### 3.4. Cardiac Magnetic Resonance

Cardiac magnetic resonance imaging (CMR) was performed in 4 patients (Figure 4). Slightly or moderately increased thickness of the interventricular septum muscle was found in 3 female carriers (12–16 mm), and significantly increased inferior wall thickness (19 mm) was observed in the male proband Arg302Gln *PRKAG2*.

It is worth noting that no late gadolinium enhancement (LGE) was detected in female carriers aged 43, 45, and 71 years, and small, patchy LGE foci were detected in a male carrier of Arg302Gln *PRKAG2* at the age of 31 years.

### 3.5. Follow-Up

After a median follow-up of 13.1 years (range 2.7–25 years), three probands with Phe293Leu, His530Arg, and Val336Leu experienced progressive HF; two of them had hypokinetic HCM. The one with the Val336Leu variant developed sustained ventricular tachycardia and received an implantable cardioverter defibrillator at the age of 32. Despite this, he died suddenly a year later during worsening of HF. In the proband with the His530Arg variant and hypokinetic HCM at first evaluation, the HF progressed with the periodic need for hospital treatment, but the course was relatively stable thanks to the escalation of pharmacological treatment and the replacement of the double-chamber pacemaker with cardiac resynchronization therapy-defibrillator at the age of 56 years. Of note, her daughter, diagnosed with HCM and massive left ventricular hypertrophy, died suddenly at the age of 15 years during physical activity. The Phe293Leu carrier, a diabetic patient, had progression of left ventricular hypertrophy with preserved LVEF, restrictive physiology, and congestive HF along with end-stage renal disease, requiring dialyses.

The female patient with Val336Leu had a myocardial infarction (MI). On coronary computed tomography angiography (CCTA) performed at the age of 71 years, no significant narrowing in the coronary arteries was found; however, bridging of the left anterior descending artery (LAD) was present (Figure 5). Following a period of increased physical activity, her follow-up CMR scan showed thickening of the heart muscle wall up to 23 mm, with a subendocardial and partly transmural focus of late gadolinium enhancement in the middle inferolateral segment, indicating fibrosis of ischemic etiology.

The Arg302Gln proband remains oligosymptomatic. He had only two SVT episodes and needs no medical therapy—antiarrhythmic treatment would probably require pacemaker implantation, as the average heart rate on Holter monitoring was 53 bpm. His mother and her sister, aged 43 and 51 years, had pacemakers implanted due to bradycardia.

## 4. Discussion

In this study, we analyzed available clinical, imaging, and observational data of patients with *PRKAG2* syndrome, carriers of four rare *PRKAG2* variants.

Current ESC standards for the management of cardiomyopathies recommend considering PS in the differential diagnosis of HCM and restrictive heart diseases [14,16]. In this study, based on published criteria [9], HCM was diagnosed in 4 probands, and 2 of the 4 identified variants, His530Arg and Val336Leu, were associated with hypokinetic HCM and HF in the end of follow-up. This is consistent with the study of Thevenon et al., who showed that among carriers of *PRKAG2* variants, a high percentage of HCM patients (25% 6/24) evolved towards severe, hypokinetic HCM [17]. Of note, this is a fundamental difference compared to classic HCM, where in patients with variants in one of the sarcomeric genes, approximately 10% develop hypokinetic HCM [17]. Interstitial fibrosis is common in HCM, and its progression over time leading to HF is well documented [18].

Comorbidities, such as long-term diabetes and chronic renal disease, could have contributed to the development of HF in the young female patient with the Phe293Leu variant [19] and concentric LVH treated also for end-stage renal disease at the end of follow-up. Only in this study did we show that PAS-positive deposits, a morphological indicator of PS, were detected in the endomyocardial biopsy of this patient who underwent a biopsy to exclude amyloidosis.

Three interesting issues are related to the Val336Leu variant, found also in a large Chinese family [20] with intrafamilial phenotypic variability and incomplete penetrance. Firstly, perhaps our most interesting discovery was related to the fact that we detected the Val336Leu *PRKAG2* variant in the patient in whom Fidzianska et al. [11] described ultrastructural changes and the presence of autophagal vacuoles in cardiomyocytes in a heart muscle biopsy taken during the transition period from HCM to HF. At that time, we did not know that our patient’s disease was caused by the *PRKAG2* variant [11]. In the study by Thevenon et al., another substitution of valine to alanine, Val336Ala *PRKAG2,* was associated with PS [17]. Of note, obstructive HCM leading to HF in the fourth decade was observed in two carriers with the Val336Ala *PRKAG2* [17], similarly to our proband with the Val336Leu variant who died suddenly at the age of 33 years.

Secondly, our proband’s mother, the Val336Leu carrier, had MI while having no significant atherosclerotic lesions; however, there was systolic compression of mid LAD on CCTA. Of interest, Borodzicz-Jażdzyk reported on a 43-year-old patient with nonobstructive HCM, a history of atrial tachycardia, and chest pain who had adenosine stress first-pass perfusion CMR [21]. The authors showed extensive perfusion defects in the hypertrophied segments of the myocardium, whereas CCTA showed no significant atherosclerotic lesions and 50–60% of LAD bridging. The authors state that the perfusion defects are highly suggestive of extensive coronary microvascular dysfunction, present in HCM [21]. This might be the case in our patient. Of interest, Acar et al., studying 227 patients with CCTA and echocardiography, found that the severity of LAD bridging was associated with LVH [22].

Thirdly, late (8th decade) penetrance was observed in the proband’s mother, with the Val336Leu variant, possibly induced by a period of increased physical activity. Ahmad et al. studied the effect of exercise in an experimental model with another *PRKAG2* Asn488Ile variant and found that it had little effect on resting cardiac metabolism but accelerated glycogen metabolism during exercise [23]. Therefore, Qiu et al. indicated that, unlike other types of cardiomyopathy, exercise should be carefully recommended to patients with *PRKAG2* cardiomyopathy [24].

The His530Arg *PRKAG2* variant, found in family 4, with a history of sudden cardiac death, was associated with serious myocardial and arrhythmic complications, as the proband had an early double chamber pacemaker implanted at the age of 21 years and developed hypokinetic HCM with subsequent HF. A similarly poor prognosis in patients with this variant was described by French researchers [17]. Of note, our variant in the position of 530 is in the vicinity of Arg531Gln, the most severe variant reported until now, which is characterized by an extreme early onset and a severe clinical course leading to death from cardiogenic shock within the first three months of life [25].

Of interest, coexistent chronic renal disease, present in two female carriers of His530Arg and Phe293Leu variants, may have a mixed etiology, as renal enlargement has been associated with another *PRKAG2* variant, Arg531Gln [25].

The Arg302Gln variant, the most commonly identified in PS worldwide, was found in one family with heterogenous presentation. The onset of the disease in our proband was paroxysmal supraventricular tachycardia, identified in a patient with short PR, a common electrocardiographic feature present in 2/7 (28.5%) carriers in our study and up to 68% of patients in the study by Gollob et al. [1]. The atrioventricular accessory pathway found on electrophysiologic study in our patient is the most commonly observed in PS [20]. Supraventricular tachyarrhythmias, particularly atrial fibrillation (AF) and atrial flutter (AFl), common among patients with PS, were reported in 38% of patients in the study by Murphy et al. [3], and Lopez-Sainz reported the presence of AF in 29% among 90 *PRKAG2* variant carriers [9]. In our study, long-standing persistent AF/AFl was found in the His530Arg variant carrier, who underwent radiofrequency ablation three times. Another essential feature of PS is conduction system disease, leading to pacemaker implantation in 21% of 90 carriers at a median age of 37 years in the study by Lopez-Sainz [9], and in 17/48 (38%) patients over a 12 year follow-up in the study by Murphy et al. [3]. Of note, in the latter study, 7 patients had pacemakers implanted for atrioventricular block and 10 for symptomatic bradycardia. This is in agreement with our study; two of our Arg302Gln carriers required pacemaker implantation for symptomatic bradycardia, and His530Arg carrier, due to the disease of both nodes (sinus and atrioventricular).

Concentric LVH is often found in myocardial storage disease, but in our patients with PS the picture was heterogeneous, as in other reports [3,9]. Although LGE on CMR study in our patients was unremarkable, as in the study by Porto et al. [26], it may be more conspicuous in later stages of the disease [9,25].

PS can be suspected based on clinical presentation, but a genetic test is necessary to confirm the diagnosis [9]. Additionally, clinical family screening may raise early suspicion of PS in the relatives of the affected patient before genetic examination is performed. In our study all *PRKAG2* variants met the ACMG criteria for pathogenicity [13].

The course of the disease in *PRKAG2* cardiomyopathy in our series seems to be related to the site of mutation in the *PRKAG2* gene, with severe complications found in carriers of *PRKAG2* Phe293Leu, His530Arg, and Val336Leu variants, and not in carriers of the Arg302Gln *PRKAG2* variant.

However, in the largest study published to date, there were no differences in cardiovascular event rates during follow-up, except for AF, which was more frequent in patients with the most common Asn488Ile and Arg302Gln *PRKAG2* variants.

In summary, PS should be suspected in young patients with LVH and conduction system disease, including sino-atrial disease and atrio-ventricular accessory pathways. It should also be considered if HCM enters the hypokinetic phase. To establish the diagnosis, molecular tests must be performed.

## Figures and Tables

**Figure 1 biomedicines-13-00751-f001:**
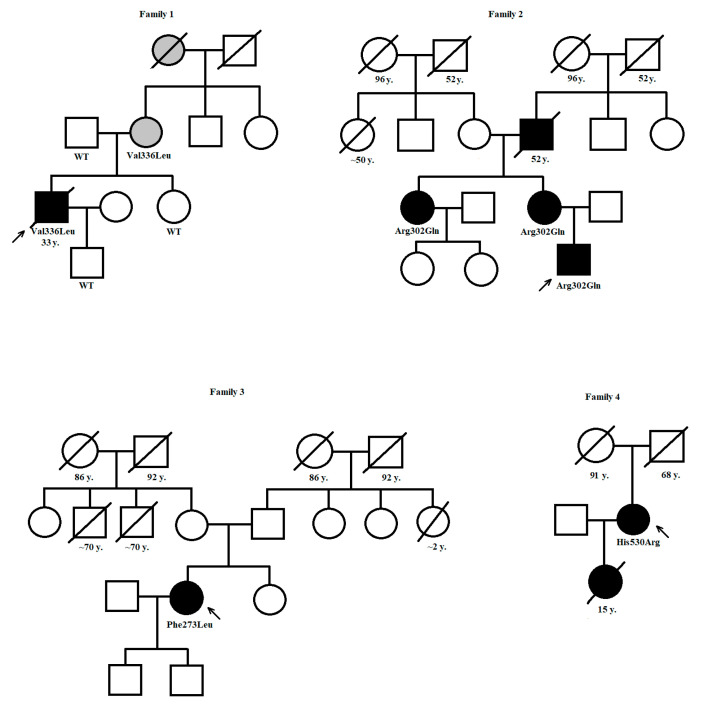
Pedigrees of families with *PRKAG2* variants. Legend: squares denote males, circles for females; numbers denote age at death in years, black color—affected, grey color—borderline/uncertain phenotype; subjects tested genetically are marked either by the name of the heterozygous *PRKAG2* variant or as “WT”, if negative for *PRKAG2* variant detected in the proband.

**Figure 2 biomedicines-13-00751-f002:**
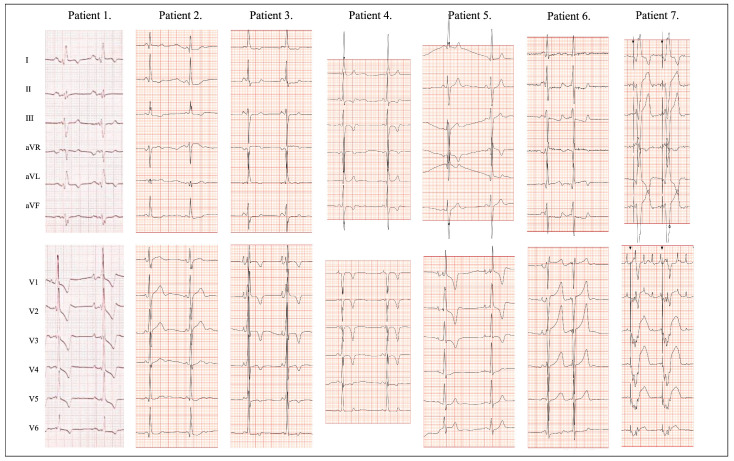
12-lead standard electrocardiogram in 7 patients with *PRKAG2* variants. Patient 1: Sinus rhythm 68 bpm, normal QRS axis of −27°, PR 194ms, QRS 150 ms, RBBB, left ventricular and right ventricular hypertrophy, biatrial enlargement, ST depression and T-wave inversions in I, aVL and V1-6 secondary to RBBB and biventricular hypertrophy. Patient 2: Sinus rhythm 62 bpm, normal QRS axis of 68°, short PR interval of 120 ms, notched QRS of 110 ms in III, aVF, V1, narrow q waves in leads V5–V6. Right ventricular hypertrophy, ST depression in II, III, aVF. Patient 3: Sinus rhythm 64 bpm, normal QRS axis of −3°, short PR interval of 113 ms, QRS 128 ms, preexcitation pattern. Patient 4: Sinus bradycardia 43 bpm, normal QRS axis of −8°, PR 127 ms, QRS 146 ms, interventricular conduction delay (LBBB-like pattern), LVH and profound repolarization abnormalities (negative deep T waves in II, III, aVF and V1–V4). Patient 5: Sinus bradycardia 46 bpm, left axis deviation of −45°, PR 150 ms, QRS 130 ms, RBBB, LAFB, left atrial enlargement, LVH, RVH. Patient 6: Sinus rhythm 60 bpm, left axis deviation of −46°, PR 140 ms, QRS 135 ms. rSR’ in V1. LVH with intraventricular conduction delay and secondary ST-T abnormalities. Biatrial enlargement. Narrow q waves in leads I, aVL, V6. Patient 7: Atypical atrial flutter 220 ms, VVI pacing 75 bpm. Shortcuts explained in the Table 2 legend.

**Figure 3 biomedicines-13-00751-f003:**
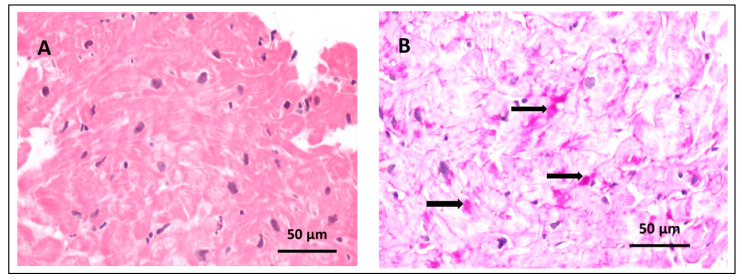
Endomyocardial bioptate images. (**A**) Hypertrophied cardiomyocytes with enlarged pleomorphic nuclei, HE ×200. (**B**) Periodic acid–Schiff–positive deposits corresponding to glycogen in cardiomyocytes (arrows), PAS ×200.

**Figure 4 biomedicines-13-00751-f004:**
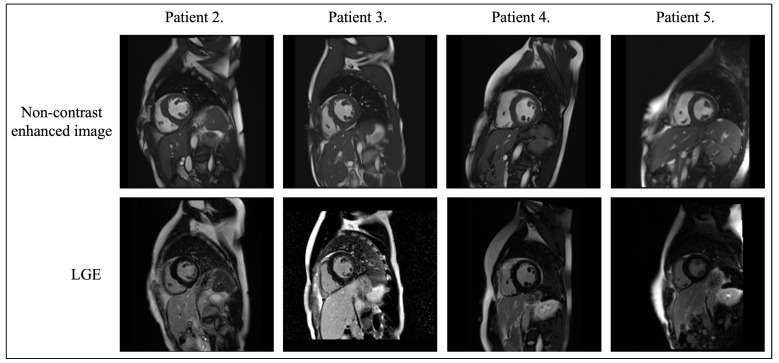
Cardiac magnetic resonance images. Patient 2: An examination performed at the age of 71 revealed increased interventricular septum thickness (16 mm), normal left ventricular (LV) function (LV ejection fraction (LVEF 75%) and enlarged left atrium. LV mass was increased (68 g/m^2^). No late gadolinium enhancement (LGE) was seen. Patient 3: A CMR study performed at the age of 31 revealed asymmetric LV hypertrophy with a maximum muscle thickness of up to 19 mm at the border of the middle segment of the inferior wall and the middle infero-septal segment. Slight enlargement of the left atrium. LVEF of 61%. The images following gadolinium-based contrast agent administration revealed small, patchy foci of LGE. Patient 4: A CMR examination performed at the age of 45 revealed the borderline for the diagnosis of cardiomyopathy thickness of the interventricular septum (12–13 mm). Good LV function (LVEF 69%) was observed. No LGE was detected. Patient 5: A CMR performed at age of 43 revealed LV wall thickness of maximum 12–13 mm within the interventricular septum and a slight enlargement of the LV cavity. Normal LV function (LVEF 77%) was preserved. No LGE was present.

**Figure 5 biomedicines-13-00751-f005:**
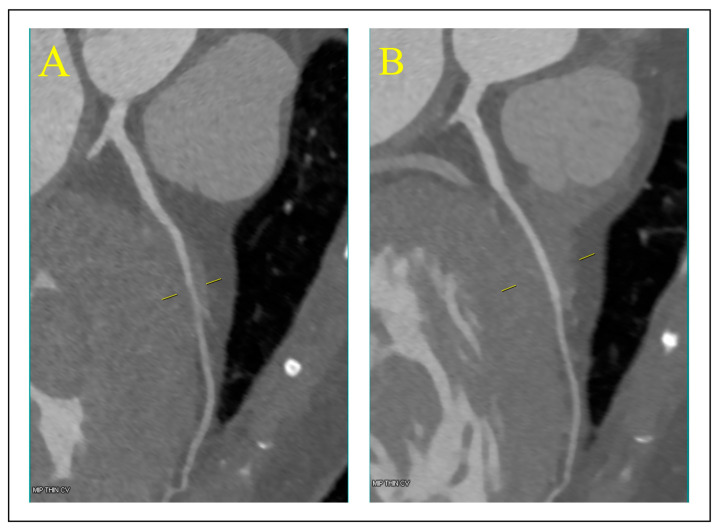
Coronary computed tomography angiography. Image of left anterior descending artery (LAD) in systole (**A**) and diastole (**B**). On the left side of the respective images coronary artery cross sections illustrating the depth of the intramuscular course of the mid arterial segment. Notably, the mid LAD within the myocardial muscle displays signs of compression with dynamic stenosis during the systolic phase.

**Table 1 biomedicines-13-00751-t001:** Characteristics of the identified *PRKAG2* variants.

Amino Acid Change (NP_077747.1)	p.Val336Leu	p.Arg302Gln	p.Phe293Leu	p.His530Arg
Nucleotide change (NM_0106203.4)	c.1006G>T	c.905G>A	c.877T>C	c.1589A>G
Coordinates (hg38)	chr7:151572709-C>A	chr7:151576412-C>T	chr7:151576440-A>G	chr7:151560613-T>C
Zygosity	heterozygous	heterozygous	heterozygous	heterozygous
MAF gnomAD v.4.1.0	0	0.000001241	0	0
MAF Million Exome Variant Browser v1.1.3	0	0.000001825	0	0
Bioinformatic score AlphaMissense	0.97 Pathogenic	1.0 Pathogenic	1.0 Pathogenic	1.0 Pathogenic
Bioinformatic score CADD	30 Pathogenic	30 Pathogenic	27 Pathogenic	25 Uncertain
Bioinformatic score REVEL	0.58 Uncertain	0.84 Pathogenic	0.89 Pathogenic	0.9 Pathogenic
ACMG classification and criteria	Likely pathogenic PM2, PM5, PP3, PP5	Pathogenic PS3, PS4, PP1, PM2, PM5, PP3	Likely pathogenic PS1, PM2, PP3	Pathogenic
ClinVar classification	Conflicting (pathogenic—1x, unknown significance—1x)	Pathogenic (20x)	-	Pathogenic (5x)
Protein domain	CBS1	CBS1	CBS1	CBS4

Table legend: ACMG—American College of Medical Genetics and Genomics, CBS—cystathionine beta-synthase, hg—Human Genome, MAF—minor allele frequency.

**Table 2 biomedicines-13-00751-t002:** Characteristics at first evaluation and follow-up data of the individuals with pathogenic *PRKAG2* variants.

	Patient	1	2	3	4	5	6	7
	*PRKAG2* variant	Val336Leu	Val336Leu	Arg302 Gln	Arg302Gln	Arg302Gln	Phe293Leu	His530Arg
	Status	roband	relative	proband	relative	relative	proband	proband
	Gender	male	female	male	female	female	female	female
Initial Evaluation	Age at onset	16 years	63 years	30 years	44 years	43 years	46 years	21 years
	Symptoms at onset	chest pain, fatigue, dizziness, palpitations	palpitations, chest pain	palpitations	dizziness	asymptomatic	chest pain, fatigue, palpitations	syncope, palpitations
	Age at first evaluation	29 years	66 years	30 years	44 years	43 years	48 years	38 years
	NYHA class	III	I	I	I	I	III	II
	Other diseases	stroke	hyperten-sion, CAD, cholelithiasis, thrombocytopenia	paroxysmal SVT	No	No	hypertension, diabetes mellitus, CKD	rheumatic fever (at age 2 and 9) CKD
Initial ECG	Rhythm	SR	SR	SR	SB	SB	SR	AFl
	PR (ms)	194	120	113	127	150	140	NA
	QRS (ms)	150	110	128	146	130	135	NA
	Preexcitation	No	Yes	Yes	No	No	No	NA
	CCD	RBBB	No	No	IVCD	RBBB, LAFB	IVCD	LBBB (earlier)
	LVH	Yes	No	No	Yes	Yes	Yes	NA
Initial ECHO	LAD (mm)	52	41	42	34	39	48	48
	LVMWT (mm)	18	11	15	13	12	24	14
	LVEF (%)	50	73	65	60	65	60	45
	RVSP (mmHg)	120	40	30	24	29	80	40
Follow-Up	Time (years)	17.9	24.7	2.7	8.2	13.1	9.4	25.0
	Outcome (age in years)	progressive HF; sudden death (33)	myocardial infarction (72), LVH progression (76)	stable (only 2 SVT episodes)	stable	stable (new hypertension)	progressive HF; CKD progression to end-stage renal failure (dialyses)	AFl ablation (30); ineffective AF ablations (48); progressive HF; coexisting CKD
	LVMWT (mm)	20	17	15	14	12	27	26
	LVEF (%)	30	60	60	67	65	73	45
	Medication	NA	ASA, BB, ACE-I, CCB, indapamide, statin	none	none	ARB, CCB	BB, ACE-I, CCB, loop diuretics, clonidine, statin, allopurinol	VKA, BB, ARNI, torasemide, MRA, SGLT2-I, statin
	CIED (age at implantation)	ICD-VR (32)	No	No	DDD (51)	DDD (43)	No	DDD (21), CRT-D (56)

Abbreviations: ACE-I—angiotensin-converting-enzyme inhibitor, AF—atrial fibrillation, AFl—atrial flutter, ARNI—angiotensin receptor neprilysin inhibitor, ASA—acetylsalicylic acid, BB—beta-blocker, CAD—coronary artery disease, CCB—calcium channel blocker, CCD—cardiac conduction disease, CIED—cardiac implantable electronic devices, CKD—chronic kidney disease, CRT-D—implantable cardiac resynchronization therapy-defibrillator, DDD—dual-chamber pacing system, ECG—electrocardiography, ECHO—echocardiography, HF—heart failure, ICD-VR—single chamber implantable cardioverter-defibrillator, IVCD—intraventricular conduction delay, LAD—left atrial diameter, LBBB—left bundle branch block, LAFB—left anterior fascicular block, LVEF—left ventricular ejection fraction, LVH—left ventricular hypertrophy, LVMWT—left ventricular maximal wall thickness, NA—not available, NYHA—New York Heart Association, RBBB—right bundle branch block, RVSP—right ventricular systolic pressure, SB—sinus bradycardia, SGLT2-I—sodium-glucose cotransporter 2 inhibitor, SR—sinus rhythm, SVT—supraventricular tachycardia, VKA—vitamin K antagonist.

## Data Availability

The data presented in this study are included in the manuscript.

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
