# Peer review of "PRKAG2* Syndrome: Clinical Features, Imaging Findings and Cardiac Events"

_biomedicines, 2025, doi:10.3390/biomedicines13030751_

Round 1
Reviewer 1 Report
Comments and Suggestions for Authors
I read with interest the article by Maria Sudomir titled: PRKAG2 syndrome. Clinical features, imaging findings and cardiac events.
I have a few comments.
This is a well-written manuscript providing some comprehensive insight into the clinical and phenotypic entity of PRKAG2 syndrome.
There is a paucity of data in literature on this very rare disease, hence, this study by providing a comprehensive FU of these 7 individuals assists in understanding this disease ever so slightly more and this is welcome.
This work is also accompanied by good quality images and graphs.
I would strongly support the publication of this work at this time.
Author Response
We are very grateful to the Reviewer for favourable review.
Reviewer 2 Report
Comments and Suggestions for Authors
The study focuses on PRKAG2 syndrome (PS), a rare but significant genocopy of hypertrophic cardiomyopathy (HCM). Given the limited literature on PS, this research expands current knowledge. Identifying a potential link between the Val336Leu PRKAG2 variant and autophagy is an important finding that may have implications for understanding the disease's pathophysiology. The study highlights early-onset arrhythmic events and their role in heart failure progression, providing valuable insights for clinicians managing patients with PRKAG2 mutations. However, there is no direct comparison with patients with classical HCM, making it challenging to determine whether the observed clinical course is unique to PS. While cardiac outcomes are well described, additional metabolic and systemic manifestations of PS (e.g., skeletal muscle involvement, and glucose metabolism abnormalities) are not explored in detail.
Author Response
We are very grateful to the Reviewer for the favourable review.
Comment 1: However, there is no direct comparison with patients with classical HCM, making it challenging to determine whether the observed clinical course is unique to PS.
Response 1: We agree that such a comparison would be interesting, but it was not the aim of this study, especially since with a group of 7 people studied over a decade it would be impossible to draw any clear conclusions.
Comment 2: While cardiac outcomes are well described, additional metabolic and systemic manifestations of PS (e.g., skeletal muscle involvement, and glucose metabolism abnormalities) are not explored in detail.
Response 2:
We did not observe skeletal muscle involvement, and severe metabolic disorders (diabetes) were observed in one patient. Both pieces of information are included in the Results section.
Reviewer 3 Report
Comments and Suggestions for Authors
In the original article 'PRKAG2 syndrome. Clinical features, imaging findings and cardiac events.' the authors identified novel PRKAG2 mutations. The topic of this manuscript is interesting. However, I suggest several points, which can/should be improved:
1.) Please write all human gene names in the complete manuscript according to the official guidelines in Italics.
2.) Please add an OMIM identifier for the PRKAG2 gene within the introduction.
3.) I would shortly summarize the genetic landscape of cardiomyopathies. For details the following book chapter ' The genetic landscape of cardiomyopathies' would be really helpful.
4.) Line 38: Please add also the c.DNA changes in brackets for each mutation
5.) Table 1: Please add the minor allele frequencies according to the Million Exome Server and the GNOMad Databases.
6.) Figure 3. Please add scale bars to this figure.
7.) Discussion: Could you discuss the impact of the mutations on the sturture and present a molecular model (e.g. by using Alphafold2/3)?
8.) Are animal models known, which could be discussed in your discussion?
9.) I would also use bioinformatic prediction tools like the Revel score or the CADD score. Please prepare a small table.
10.) In addition, which other variants were identified. Please prepare for each patient an additional table and explain why these other genetic variants were excluded as disease causing variants.
11.) I would add a gene list in the supplements indicating which genes were included in the sequencing panels?
12.) Do you have verified the identified mutations by using Sanger sequencing?
In summary, I suggest a major revision for this manuscript. However, I am pretty optimistic, that the authors can fix the critized points.
Author Response
We thank the Reviewer for insightful review. We have answered all Reviewers questions/comments. The results are presented below:
- Please write all human gene names in the complete manuscript according to the official guidelines in Italics. -corrected in the manuscript
2.) Please add an OMIM identifier for the PRKAG2 gene within the introduction. – OMIM entry 602743- added in the introduction .
3.) I would shortly summarize the genetic landscape of cardiomyopathies. For details the following
book chapter ' The genetic landscape of cardiomyopathies' would be really helpful.
Few sentences summarizing genetics of HCM and their genocopies added in the introduction.
4.) Line 38: Please add also the c.DNA changes in brackets for each mutation – unfortunately due to
250 word limit in the abstract we cannot do it – full name with c.DNA level in Table 1
5.) Table 1: Please add the minor allele frequencies according to the Million Exome Server and the
GNOMad Databases – MAF frequencies added to the Table 1 .
6) Figure 3. Please add scale bars to this figure. - done
7.) Discussion: Could you discuss the impact of the mutations on the structure and present a molecular model (e.g. by using Alphafold2/3)? Unfortunately, in this team of authors we do not have the expertise necessary to carry out these analyses, at the same time we emphasize that this work is of a clinical nature.
8.) Are animal models known, which could be discussed in your discussion? “Several transgenic murine models of PRKAG2 variants were developed. They all uniformly reflected the disease observed in humans, with cardiac hypertrophy, dilatation and decreased contractility of the left ventricle found in echocardiography, and pre-excitation features in surface ECG” – added in the intrudction.
9.) I would also use bioinformatic prediction tools like the Revel score or the CADD score. Please
prepare a small table – due to the fact that several bioinformatics scores are already included in ACMG PP3 criterion we added 3 rows to Table 1 with bioinformatics scores from: alphamissense, CADD and Revel and ACMG criteria triggered. We hope this information will be sufficient.
10.) In addition, which other variants were identified. Please prepare for each patient an additional table and explain why these other genetic variants were excluded as disease causing variants.
No other protein-coding or splice site variants with MAF <0.001 were found in our probands in HCM-associated genes.– this information added to the Results section.
The following genes were analysed: ACTC1, ACTN2, ALPK3,CACNA1C, CSRP3, DES, FHL1, FHOD3, FLNC, GLA, JPH2, KLHL24, LAMP2, MYBPC3, MYH7, MYL2, MYL3, MT-TI, PLN, PTPN11, RAF1, RIT1, TNNC1, TNNI3, TNNT2, TPM1, TRIM63, TTR.
11.) I would add a gene list in the supplements indicating which genes were included in the sequencing panels?
We used commercial panels, WES panel contained >20 000 protein coding genes, TSO >4800 genes, TSC 174 genes – list of genes for this commercial available panels are available on producers web pages.
12.) Do you have verified the identified mutations by using Sanger sequencing?
Yes– information was included in Materials and Methods section – Genetic testing
Round 2
Reviewer 3 Report
Comments and Suggestions for Authors
From my perspective the authors have significantly improved their manuscript and have addressed the points, which I have criticized before.